# Health-Related Quality of Life and Mental Well-Being during the COVID-19 Pandemic in Five Countries: A One-Year Longitudinal Study

**DOI:** 10.3390/jcm11216467

**Published:** 2022-10-31

**Authors:** Di Long, Gouke J. Bonsel, Erica I. Lubetkin, John N. Yfantopoulos, Mathieu F. Janssen, Juanita A. Haagsma

**Affiliations:** 1Department of Public Health, Erasmus MC, 3000 CA Rotterdam, The Netherlands; 2EuroQol Research Foundation, 3068 AV Rotterdam, The Netherlands; 3Department of Community Health and Social Medicine, CUNY School of Medicine, New York, NY 10031, USA; 4Health Department of Economics, National and Kapodistrian University of Athens, 157 72 Athens, Greece; 5Section Medical Psychology and Psychotherapy, Department of Psychiatry, Erasmus MC, 3000 CA Rotterdam, The Netherlands

**Keywords:** health-related quality of Life, mental well-being, EQ-5D-5L, WHO-5, longitudinal, COVID-19

## Abstract

This study aimed to investigate socioeconomic and health-related determinants and recent life events and their relation to changes in health-related quality of life (HRQoL) and mental well-being during the first year of the COVID-19 pandemic. A web-based survey was administered repeatedly to participants from Greece, Italy, the Netherlands, the United Kingdom, and the United States. Primary outcome measures were HRQoL (measured by EQ-5D-5L) and mental well-being (measured by WHO-5). Linear regression analyses were performed to estimate the impact of determinants on HRQoL and well-being. In total, 6765 respondents completed the questionnaire at T1 (April–May 2020) and T2 (May–June 2021). Regarding results, 33% showed improved HRQoL at T2, whereas 31% deteriorated. In terms of mental well-being, 44% improved and 41% deteriorated. The greatest deterioration in HRQoL and mental well-being from T1 to T2 was observed with an increasing number of chronic conditions. The effect of negative life events on HRQoL and mental well-being was larger than the effect of positive life events. We conclude that slightly more respondents showed improved rather than deteriorated HRQoL and mental well-being, with some variation by outcome measure and country.

## 1. Introduction

Two years after its onset, the COVID-19 pandemic is far from over due to insufficient containment measures, stagnating vaccination coverage, lack of resources, and new virus variants emerging. Consequences are noticeable on the global economy [1,2], at the health care level [3,4], and in the daily lives of individuals [5,6], going beyond the impact on those acutely infected.

Many studies indicated that the COVID-19 pandemic is related to various health problems in the general population. These health problems cover a range of health dimensions, such as physical symptoms and sequelae [7], psychological problems [8], social functioning [9,10], sleep [11], and fatigue [12].

Since the first wave of the pandemic, the patterns of incidence of COVID-19 infection and government responses varied across countries, with vaccination as an important modifier [13]. While the focus of health care is usually on treatment of acute infection and post-infection health states, the indirect impact of the COVID-19 pandemic on the general population in terms of health-related quality of life (HRQoL) and mental well-being may outsize the impact of those individuals infected. HRQoL, defined as the degree to which a person functions in their life, is a multidimensional concept and reflects their perceived well-being in physical, mental, and social domains of health [14]. Mental well-being is defined by the World Health Organization (WHO) as “a state of well-being in which the individual realized his or her own abilities, can cope with the normal stresses of life, can work productively and fruitfully, and is able to make a contribution to his or her community” [15].

The impact of the pandemic on HRQoL and mental well-being varies across region and according to demographic and epidemiological characteristics. Apart from country of residence and local restriction measures against the spread of the COVID-19 virus, several determinants contribute to HRQoL and mental well-being differences, such as pre-existing chronic illness, age and gender, socioeconomic status, race and ethnicity, and negative recent life events related to determinant changes such as job loss and loss of health insurance [16,17,18,19,20,21,22,23,24].

In addition, as symptoms after the acute phase of COVID-19 infection–often referred to as “long COVID”–may persist long after acute COVID-19 [25], the noticeable presence of long COVID may impact the trajectory of HRQoL and mental well-being changes in the general population.

Information on the longitudinal evolution of HRQoL and mental well-being could indicate which vulnerable groups need to be monitored more closely in the post-COVID era and which potentially modifiable factors must be addressed in the frail population. Few studies have reported HRQoL and mental well-being longitudinally in the general population during the pandemic on a multi-country level. Such studies were small or focused on a single country [26,27,28,29,30], unable to detect the balance of impacts in disadvantaged groups.

Therefore, the aims of this study were twofold. First, we investigated how HRQoL and mental well-being developed one year after the first wave of the pandemic. Second, we explored socioeconomic and health-related determinants and recent life events related to health, work, income, and living situation and their relation to changes in HRQoL and mental well-being over the follow-up period.

## 2. Materials and Methods

### 2.1. Study Design and Population

This study is part of the second wave of the POPulation health impact of the CORoNavirus disease 2019 (COVID-19) pandemic (POPCORN) study. The POPCORN study is a study that commenced in April 2020 that aims to measure the impact of the COVID-19 pandemic on health-related quality of life in the general population of multiple countries. In this longitudinal study, a web-based survey was administered to the participants at T1 (between 22 April and 5 May 2020) and T2 (between 3 May to 29 June 2021) in five countries: Greece, Italy, the Netherlands, the United Kingdom (UK), and the United States (US). Detailed results of T1 data collection have been published elsewhere [31].

### 2.2. Data Collection Procedure and Consent

The participants were recruited by an international market research agency (Dynata) that distributed and launched the questionnaire. A representative sample by age and sex in several countries was invited to participate in the first questionnaire at T1, and the respondents that completed the first questionnaire at T1 were then invited to fill out the questionnaire again at T2. The participants were members of the market research agency’s existing voluntary panels. As panel members, the participants had already provided written informed consent to participate in online surveys upon registration. Once participation had begun, the data capture system did not allow for the skipping of questions or for questions to be left unanswered. As a result, there were no missing values. Participants received an incentive in the form of cash or points from the market research company upon completion of the survey. Data were anonymized.

The questionnaire was translated into the main official language of each country using translation software and subsequently translated back into English, except when validated translated versions of the instruments were available. Bilingual native speakers verified the translations independently.

### 2.3. Primary Outcome Measures

The questionnaire included items on demographic and socioeconomic determinants, health-related and COVID-19-related determinants, the EQ-5D-5L and World Health Organization-Five Well-being (WHO-5) instruments, and other related instruments (see also [30]). We defined change in HRQoL (measured by the EQ-5D-5L) and change in mental well-being (measured by the WHO-5) as the primary outcomes in this study. Since these outcome measures provide numerical scores, change was calculated as the score difference between T2 and T1 (T2 minus T1) for each outcome measure, which can have a positive (improvement) or negative (deterioration) sign.

The EQ-5D-5L assesses five dimensions of HRQoL and the reference period is “today” [32]. The EQ-5D-5L consists of five dimensions: Mobility, Self-Care, Usual Activities, Pain/Discomfort, and Anxiety/Depression. The ordinal response options range from “No problems” (“1”) to “Extreme problems/Unable to (“5”). The EQ-5D-5L level sum score is the summary of the scores of each dimension and ranges from 5 (best) to 25 (worst). The EQ-5D-5L index is a weighted sum of the level scores using a value set (i.e., five dimension-specific weights), which reflects societal preferences for EQ-5D-5L health states for a country-specific population [33]. As value sets are country-specific, for comparative purposes we selected the US value set [34] to be used for all countries. The EQ-5D-5L (US) index ranges from −0.573 (all dimensions at worst level 5) to 1 (“Full health”), where 0 reflects the threshold to states considered worse than dead. The EQ VAS (EQ visual analogue scale), as part of the EQ-5D-5L instrument, is a self-rated visual analog scale assessing the respondents’ health state today. It ranges from 0 (“The worst imaginable health state”) to 100 (“The best imaginable health state”).

The WHO-5 measures mental well-being in the past two weeks [35]. The WHO-5 consists of five items: “I have felt cheerful and in good spirits”, “I have felt calm and relaxed”, “I have felt active and rigorous”, “I woke up feeling fresh and rested”, and “My daily life has been filled with things that interest me”. The ordinal response options range from “All the time” (“5”) to “At no time” (“0”). The WHO-5 index is the summary of the scores of each item multiplied by 4 and all items have the same weight. It ranges from 0 (“Worst imaginable mental well-being”) to 100 (“Best imaginable mental well-being”).

### 2.4. Respondent Characteristics at T1

Information was collected on age, sex, highest level of education achieved, income, occupational status, chronic disease status, COVID-19 status, COVID-19 vaccination status, and living situation. The highest level of education achieved is categorized into three groups according to the International Standard Classification of Education (ISCED) 2011: ISCED 0–2 (“Low”), ISCED 3–4 (“Middle”), and ISCED 5–8 (“High”). Two income variables were collected: monthly personal income (Greece) and annual household income (all other countries). Income was categorized into four groups: lower 20% (“Low”), middle 60% (“Middle”), higher 20% (“High”), and “Prefer not to answer”. Chronic disease status was measured by the presence of up to 11 chronic conditions (asthma or chronic bronchitis, heart disease, stroke, diabetes, arthritis, severe back complaints, arthrosis, cancer, memory problems, depression or anxiety disorder, and/or other problems). The number of chronic diseases was categorized into five groups: “Zero”, “One”, “Two”, “Three”, and “Four or more”.

### 2.5. Life Events Related to Health, Work, Income, and Living Situation

For the purpose of this study, changes in determinants were also included, namely changes in the number of chronic conditions, COVID-19 status, occupation status, and living situation. These determinants were measured at both T1 and T2 and the changes were captured by recording the difference between T1 and T2, with the determinants then re-categorized into meaningful groups that were referred to as “recent life event”. Change in number of chronic conditions was categorized into three groups: “Decreased”, “Same”, and “Increased”. Change in COVID-19 status was categorized into three groups: “No (past) COVID-19 infection at T1 and T2”, “(past) COVID-19 infection at/before T1”, and “(past) COVID-19 infection between T1 and T2”. Change in occupation status was assessed only for those who were (self-)employed or unemployed at T1 and was categorized into four groups: “Gained job”, “Kept job”, “Lost job”, and “Remained unemployed”. Change in living situation was categorized into “Not living alone at T1 and T2”, “Living alone at T1 and not living alone at T2”, “Not living alone at T1 and living alone at T2”, and “Living alone at T1 and T2”. Change in household income in the past year was assessed with a separate question with answering options of “Improved”, “Remained the same”, “Decreased”, and “Don’t know”. COVID-19 vaccination was not yet approved or available during T1 data collection. Consequently, change in COVID-19 vaccination was categorized into two groups: “Received COVID-19 vaccine” and “Did not receive COVID-19 vaccine”.

### 2.6. Statistical Analysis

Descriptive analyses were performed for sociodemographic data, EQ-5D-5L dimensions, EQ-5D-5L level sum score, EQ-5D-5L index, EQ VAS, and WHO-5 items in each country. A nonresponse analysis of sociodemographic data was performed to study whether T2 participants differed from T2 nonparticipants. Numerical variables were compared using Kruskal–Wallis tests. Categorical variables were compared using Chi-square tests.

For analysis, the EQ-5D-5L level sum score and EQ-5D-5L index were transformed to a 0–100 scale (with 0 as the worse anchor) in order to be comparable with the EQ VAS and WHO-5 index [36]. Formulas were:Transformed EQ-5D-5L level sum score (tEQ-5D-5L level sum score) = [(EQ-5D-5L level sum score − 25) × (−5)];
Transformed EQ-5D-5L index score (tEQ-5D-5L index score) = (EQ-5D-5L index score × 100).

We then computed the difference score (“change”) by subtracting the T1 score from the T2 score for each primary outcome measure. For some analyses, we converted the T1–T2 change of the primary outcome measures into improved (score T1 < score T2), no change (score T1 = score T2), and deteriorated (score T1 > score T2).

Change in primary outcome measure was graphically displayed by Sankey plots. Univariate linear regression analysis was applied to assess which factors predict change (T2 minus T1) in tEQ-5D-5L level sum score, tEQ-5D-5L index, EQ VAS, and WHO-5 index. For each outcome, univariate regression analysis was repeated for each baseline (T1) determinant as an independent variable.

Next, for each outcome measure (change in tEQ-5D-5L level sum score, tEQ-5D-5L index, EQ VAS, and WHO-5 index), multivariate stepwise regression analysis (using backwards elimination) was performed using baseline (T1) determinants as independent variables, including sex, age, highest level of education achieved, occupational status, income, chronic disease status, COVID-19 status, and living situation as potential explanatory variables, as well as recent life events and country of residency. Backwards elimination was used until only significant variables remained. Multivariable regression coefficient estimates (betas) were reported in tables. Data on personal income were missing from 12 respondents residing in Greece. We therefore excluded data from these participants from the regression analysis.

We calculated the mean change in tEQ-5D-5L level sum score, tEQ-5D-5L index, EQ VAS, and WHO-5 index by life event category. We tested for differences in the changes in tEQ-5D-5L level sum score, tEQ-5D-5L index, EQ VAS, and WHO-5 index by life event category with the independent-samples Kruskal–Wallis test.

Statistical significance was determined by a value of *p* < 0.05.

All statistical analyses were carried out using R version 4.0.5 and SPSS version 25 for Windows (IBM SPSS Statistics, SPSS Inc., Chicago, IL, USA).

## 3. Results

### 3.1. Study Population

Out of the 16,683 respondents that completed the questionnaire at T1, 6765 (response rate: 41%) completed the questionnaire at T2. The response rate ranged from 32% among the US respondents to 56% among Italian respondents. Table 1 shows the baseline characteristics at T1 and changes from T1 to T2 among the 6765 respondents. At T1, the median (IQR) age of all respondents was 56 (20). Slightly more than half of all respondents were female (52%), high-educated (55%), or without chronic conditions (57%). Results of the nonresponse analysis can be found in Appendix A
Table A1.

### 3.2. Changes in HRQoL and Mental Well-Being between T1 and T2

The mean EQ-5D-5L level sum score, EQ-5D-5L index, EQ VAS, and WHO-5 index score at T1 are shown in Appendix A
Table A2. Figure 1A,B depict the score differences for EQ-5D-5L dimensions, EQ-5D-5L level sum score, EQ-5D-5L index, EQ VAS, and WHO-5 index for all respondents. Of all respondents, going from T1 to T2, 65% to 91% reported the same EQ-5D-5L dimension scores (Figure 1). Changes in scores were most prevalent in the Anxiety/Depression dimension, with 21% improving and 14% deteriorating. About one third of respondents showed improved HRQoL scores at T2 (tEQ-5D-5L level sum score: 31% improved; tEQ-5D-5L index: 33% improved; and EQ VAS: 32% improved), whereas 29–41% deteriorated (tEQ-5D-5L level sum score: 29% deteriorated; tEQ-5D-5L index: 31% deteriorated; and EQ VAS: 39% deteriorated).

Country-specific analysis showed HRQoL improvement to be most common in Greece (tEQ-5D-5L level sum score: 38% improved; tEQ-5D-5L index: 40% improved; and EQ VAS: 35% improved), whereas deterioration was most prevalent in the UK (tEQ-5D-5L level sum score: 31% deteriorated; tEQ-5D-5L index: 34% deteriorated; and EQ VAS: 43% deteriorated).

In terms of mental well-being, 44% improved, 16% showed no difference, while 41% reported deterioration. Improvement of WHO-5 index occurred most frequently in the United States (50%), whereas deterioration was most frequent among respondents residing in Greece (54%).

### 3.3. Determinants of Change in HRQoL and Mental Well-Being

Table 2 shows the univariate regression outcomes for the change in tEQ-5D-5L level sum score, tEQ-5D-5L index, EQ VAS, and WHO-5 index. Negative coefficients represent a greater deterioration compared to the reference group. Compared to respondents with a high income, low income levels were associated with a greater deterioration in HRQoL as measured by tEQ-5D-5L level sum score, tEQ-5D-5L index, and EQ VAS. In addition, compared to respondents who lived with others, living alone was associated with a greater deterioration in HRQoL as measured by the tEQ-5D-5L level sum score and tEQ-5D-5L index. By contrast, compared to their healthy counterparts, having one or more chronic diseases and (past) infection with COVID-19 at T1 were associated with a greater improvement in HRQoL as measured by the tEQ-5D-5L level sum score and tEQ-5D-5L index.

Compared to respondents with a high education level, a low education level was associated with a greater deterioration in mental well-being, as measured by the WHO-5 index. Being a student, having one or more chronic conditions, and (past) COVID-19 infection at T1 were associated with a greater improvement in mental well-being, as measured by the WHO-5 index.

Table 3 shows the results of the multivariable analyses. After controlling for other factors, (past/early) COVID-19 infection at T1 was consistently associated with improved HRQoL (as measured by the tEQ-5D-5L level sum score, tEQ-5D-5L index, and EQ VAS) and improved mental well-being (as measured by the WHO-5 index). Other factors that were independently associated with improved HRQoL were older age (tEQ-5D-5L level sum score) and having one or more chronic conditions (tEQ-5D-5L level sum score and tEQ-5D-5L index). Factors associated with greater deterioration of HRQoL were lower income level (tEQ-5D-5L level sum score and tEQ-5D-5L index) and living alone (tEQ-5D-5L index). Lower educational level was independently associated with deteriorated mental well-being.

Association between life events and changes in HRQoL and mental well-being.

Most respondents experienced no change in terms of the number of chronic conditions (69%): 13% reported more chronic conditions at T2 and 18% less. At T1, 98% of all respondents reported no (past) COVID-19 infection, whereas at T2 this percentage decreased to 95%. At T2, the majority of the respondents (58%) reported being vaccinated.

With regards to household income, 67% of respondents experienced no changes (12% had a higher income at T2; 20% had a lower income at T2). Of the respondents who were (self-)employed at T1, 97% had kept their job and 3% had lost their job. Of those who were unemployed at T1, 66% had remained unemployed and 34% had gained employment.

The number of chronic conditions, and changes thereof, had the highest impact on all health outcomes (Table 4). Additionally, (past) COVID-19 infection at T2 (and not T1) and lower income at T2 resulted, on average, in the greatest deterioration measured by tEQ-5D-5L level sum scores. In terms of mental well-being, losing a job and lower income at T2 resulted, on average, in the greatest deterioration. On average, the effect of negative recent life events on HRQoL and mental well-being was larger than the effect of positive recent life events (in absolute terms).

## 4. Discussion

### 4.1. Summary of Main Findings

The majority of the respondents reported the same EQ-5D-5L dimension scores at T1 and T2. Nevertheless, slightly more respondents showed improved rather than deteriorated HRQoL and mental well-being, with some variation by outcome measure and country. Respondents residing in the US experienced the most deterioration in HRQoL but the largest improvement in mental well-being, whereas the opposite was found for respondents residing in Greece. In addition, in this study, positive recent life events related to health, income, and work appear to be related to improved HRQoL and mental well-being, as well as the reverse. The effect of negative recent life events was, however, larger than the effect of positive recent life events (in absolute terms).

### 4.2. Interpretation

Our study highlights the importance of measuring HRQoL longitudinally during the pandemic. We found no difference to slight deterioration in HRQoL among most of the respondents in different countries in our study. Many studies have suggested that HRQoL and mental well-being during the pandemic in the general population are persistently poorer compared to before the pandemic [29,37,38,39], although in certain sub-populations no changes were observed [40]. Our study results may indicate that the effects of the pandemic have persisted, even among those that have not been infected by COVID-19. However, it should be noted that pre-COVID-19 HRQoL and mental well-being data are not available for the majority of respondents.

Our study found opposite patterns for mental well-being and HRQoL among respondents from different countries. Mental well-being deteriorated among respondents residing in Greece, the Netherlands, and the UK, where, on average, HRQoL did not change; contrastingly, mental well-being improved in Italy and the US, where no changes or deterioration in HRQoL were found. The opposing patterns of HRQoL and mental well-being over time and across countries could be due to differences in the epidemiological profile of COVID-19 infections as well as stringency in government measures against the spread of COVID-19 that varied over time and across countries during the course of the pandemic [41,42]. Variation in patterns of mental well-being between countries may also be explained by differences in the pace of COVID vaccine administration among the general population across countries, particularly since COVID vaccination has been associated with decreases in pandemic-related distress and perceived risk of infection [43]. Furthermore, cross country differences in the share of jobs at risk due to COVID-19 [44], governmental policy regarding financial support to citizens and vulnerable businesses during the COVID-19 pandemic, and satisfaction with government responses to COVID-19 [45,46] may play a role in the opposite patterns of mental well-being and HRQoL among respondents from different countries.

Furthermore, lifestyle changes during home confinement and limited access to health care for treatment of chronic diseases may have played a role [47,48,49,50]. The impact of COVID-19 infections, government measures, lifestyle changes, and limited access to health care on the physical domain may persist [51], while the impact on the mental domain may be slightly reduced because of mental adaptation [52]. One study has found that exposure to major stressful events can lead to large short-term effects on mental health, but mental adaptation to these major events often occurs gradually and mental health may even return to pre-event levels [53]. This is also reflected by the Anxiety/Depression dimension results from the EQ-5D-5L, which showed a similar pattern to mental well-being measured by the WHO-5 index.

Our study found significant relations between socioeconomic and health determinants and HRQoL and mental well-being changes. Moreover, we found that positive and negative recent life events related to health, income, and work significantly impacted HRQoL and mental well-being changes, and the impact of negative recent life events was larger than the positive events. These findings are in line with other studies, which found negative recent life events to have both short-term and long-term impact on people’s well-being [54], with the impact also often larger than positive life events [55]. In general, mechanisms of (positive or negative) adaptation, anticipation, and selection provide a buffer against the impact of life events on health outcomes, while the rate of these mechanisms varied considerably between different life events [55,56]. However, while other studies have consistently reported on the relation between negative life events and ill health (physical and mental), contradictory findings have been reported regarding positive life events [57]. Factors such as self-esteem moderate the relation between positive life events and health outcomes [58]. Furthermore, our findings may suggest that interventions that prevent or limit negative life events, and subsequently negative changes in HRQoL and mental well-being in the general population, are most likely to be more effective than interventions that enhance positive life events or changes. However, people with few resources or those marginalized due to their socioeconomic status are experiencing a greater negative impact from the pandemic with fewer positive recent life events [59]. In turn, these populations may become more vulnerable due to low overall health and smaller resultant improvements in HRQoL and mental well-being.

Reporting an increased number of chronic conditions during the pandemic yielded the most negative effect on HRQoL and mental well-being. This is in line with previous studies that found a negative relationship between an increasing number of chronic diseases and HRQoL [60,61]. During the pandemic, care for chronic conditions was often postponed, cancelled, or neglected due to the prioritization of acute COVID patients [62]. This may have resulted in an additional toll on the HRQoL and mental well-being of persons living with chronic disease.

In contrast to other studies, we did not find associations between gender or age and significant improvement or deterioration in mental well-being in our study. With regards to gender, several studies have found differences in patterns of mental well-being during the pandemic between males and females, with males showing a more stable pattern of mental well-being over time [63,64,65,66,67,68]. With regards to age, the observed patterns of change in mental well-being during the course of the COVID-19 pandemic across studies was less consistent. For instance, Choi et al., Vistisen et al., and Pierce et al. found that, during the COVID-19 pandemic, the mental well-being of older persons changed less compared to younger people and younger age was associated with deteriorated mental well-being [66,68,69]. Contrastingly, Pieh et al., while also observing a more stable pattern of mental well-being among older persons, found that younger age was associated with improved mental well-being during the course of the pandemic [70].

The contradictory findings from our study and other studies may be explained by differences in the instrument(s) used to measure mental well-being, differences in the country of residence for the participants of the various studies, and differences in the time period in which the data on mental well-being were collected, since stringency of governmental measures against the spread of COVID-19 varied over time and across countries.

HRQoL and mental well-being changes were similar for respondents who reported a COVID-19 infection between T1 and T2 and for those who reported no COVID-19 infection. This may possibly be due to the protective effect of COVID-19 vaccines, which reduced the severity of COVID-19 symptoms [71]. Our results further found an association between the protective effect of vaccines and mental well-being, which is consistent with other studies [43,72]. Nevertheless, with the possible long-term consequence of COVID-19, any chronic conditions that are caused by COVID-19 have the potential to impair the HRQoL and mental well-being of the general population [73].

### 4.3. Strengths and Limitations

This study is one of the first large multi-country longitudinal studies to assess HRQoL changes during the COVID-19 pandemic. The first wave of data were collected during the early phase of the COVID-19 pandemic, allowing for the comparison of HRQoL and mental well-being during times of acute disruption to daily life, social isolation, and the discontinuation of usual health care, as well as for comparison to later phases of the pandemic.

There are several limitations to the study. First, respondents that did not fill out the questionnaire at T2 were significantly younger and more often reported having chronic conditions. Previous studies have suggested that these sub-groups have a higher risk of poorer health, especially in terms of the Anxiety/Depression dimension [74]. Therefore, we might have missed relevant groups that might have experienced larger changes in HRQoL and mental well-being. Second, even though our sample at T1 was representative of the general population by age and sex, participants who were more highly educated were over-represented. This lack of representativeness might underestimate HRQoL and mental well-being changes, given that persons with lower levels of education have been noted to be more vulnerable to worse HRQoL and mental well-being during the pandemic [75]. Third, in different countries, HRQoL and mental well-being changes may follow pandemic trajectories, with different COVID-19 incidence, restrictions, and vaccination distributions. Therefore, using only two measurements may only have partially captured the changes in HRQoL and mental well-being during the pandemic. Frequent estimations may offer a better image of pattern changes [76].

## 5. Conclusions

Our multi-country study on the course of HRQoL and mental well-being from the first wave of COVID-19 showed that slightly more respondents showed improved rather than deteriorated HRQoL and mental well-being, with some variation by outcome measure and country. The effect of negative life events on HRQoL and mental well-being was larger than the effect of positive life events.

## Figures and Tables

**Figure 1 jcm-11-06467-f001:**
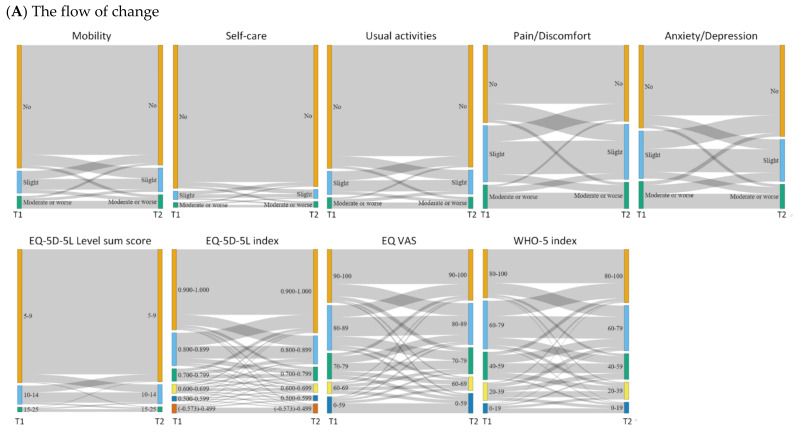
Changes in EQ-5D-5L scores (dimensions, level sum score, index, and EQ VAS) and WHO–5 index between T2 and T1 in all countries (N = 6765). Note to Figure 1B: Same: score at T1 = score T2; Better at T2: better health at T2 (T2 score >T1 if it was a positive score); Worse a T2: worse health at T2 (T2 score < T1). “ALL”, “GR”, “IT”, “NL”, “UK”, “US” represents “All countries”, “Greece”, “Italy”, “The Netherlands”, “The UK”, and “The US”, respectively.

**Table 1 jcm-11-06467-t001:** Characteristics of respondents at T1 (April–May 2020).

	Greece	Italy	Netherlands	UK	US	Total
Number of respondents at T2	511	1784	1143	1448	1879	6765
Response rate (T2/T1)	50%	56%	35%	45%	32%	35%
Baseline characteristics (T1)						
Age						
Median (IQR)	43.3 (12.5)	46.8 (13.7)	54.1 (13.9)	51.5 (14.2)	54.3 (13.0)	50.8 (14.1)
Mean (SD)	43 (18)	45 (21)	57 (21)	53 (24)	56 (20)	51 (23)
Age groups						
18–24 yrs.	43 (8%)	74 (4%)	40 (3%)	47 (3%)	25 (1%)	229 (3%)
25–34 yrs.	91 (18%)	296 (17%)	84 (7%)	171 (12%)	142 (8%)	784 (12%)
35–44 yrs.	135 (26%)	464 (26%)	172 (15%)	273 (19%)	287 (15%)	1331 (20%)
45–54 yrs.	128 (25%)	413 (23%)	227 (20%)	281 (19%)	425 (23%)	1474 (22%)
55–64 yrs.	88 (17%)	291 (16%)	295 (26%)	344 (24%)	498 (27%)	1516 (22%)
65–75 yrs.	26 (5%)	246 (14%)	325 (28%)	332 (23%)	502 (27%)	1431 (21%)
Sex						
Male	263 (51%)	859 (48%)	526 (46%)	709 (49%)	869 (46%)	3226 (48%)
Female	248 (49%)	925 (52%)	617 (54%)	739 (51%)	1010 (54%)	3539 (52%)
Education level						
High	343 (67%)	726 (41%)	464 (41%)	807 (56%)	1393 (74%)	3733 (55%)
Middle	153 (30%)	786 (44%)	351 (31%)	608 (42%)	434 (23%)	2332 (34%)
Low	15 (3%)	272 (15%)	328 (29%)	33 (2%)	52 (3%)	700 (10%)
Occupation status						
Employed	289 (57%)	1008 (57%)	552 (48%)	797 (55%)	976 (52%)	3622 (54%)
Student	31 (6%)	77 (4%)	34 (3%)	17 (1%)	15 (1%)	174 (3%)
Unemployed	134 (26%)	415 (23%)	124 (11%)	160 (11%)	234 (12%)	1067 (16%)
Retired	52 (10%)	269 (15%)	305 (27%)	365 (25%)	559 (30%)	1550 (23%)
Unable to work	5 (1%)	15 (1%)	128 (11%)	109 (8%)	95 (5%)	352 (5%)
Income level						
High	177 (35%)	227 (13%)	215 (19%)	327 (23%)	490 (26%)	1448 (21%)
Middle	162 (32%)	998 (56%)	513 (45%)	616 (43%)	986 (52%)	1879 (48%)
Low	121 (24%)	369 (21%)	182 (16%)	383 (26%)	290 (15%)	1143 (15%)
Unwilling to tell	27 (5%)	190 (11%)	233 (20%)	122 (8%)	113 (6%)	1784 (6%)
Unknown	12 (3%)	-	-	-	-	-
Number of chronic conditions						
0	304 (59%)	1128 (63%)	545 (48%)	830 (57%)	1081 (58%)	3888 (57%)
1	153 (30%)	461 (26%)	369 (32%)	379 (26%)	508 (27%)	1870 (28%)
2	36 (7%)	118 (7%)	139 (12%)	152 (10%)	181 (10%)	626 (9%)
3	11 (2%)	41 (2%)	54 (5%)	57 (4%)	67 (4%)	230 (3%)
4 or more	7 (1%)	36 (2%)	36 (3%)	30 (2%)	42 (2%)	151 (2%)
COVID-19 status at T1						
Not infected	507 (99%)	1756 (98%)	1115 (98%)	1421 (98%)	1823 (97%)	6662 (98%)
Infected	4 (1%)	28 (2%)	28 (2%)	157 (2%)	56 (3%)	143 (2%)
Living situation						
Not living alone	437 (86%)	1615 (91%)	810 (71%)	1143 (79%)	1472 (78%)	5477 (81%)
Living alone	74 (14%)	169 (9%)	333 (29%)	305 (21%)	407 (22%)	1288 (19%)

**Table 2 jcm-11-06467-t002:** Univariate analyses of respondent characteristics and the change in EQ-5D-5L transformed level sum scores, EQ-5D-5L transformed index, EQ VAS, and WHO-5 index.

	Change between T1 (April–May 2020) and T2 (May–June 2021)								
Characteristic	N	tEQ-5D Level Sum Score		tEQ-5D Index		EQ VAS		WHO-5 Index	
		Coefficient	*p*-Value	Coefficient	*p*-Value	Coefficient	*p*-Value	Coefficient	*p*-Value
Age group									
Intercept		0.1	0.894	0.5	0.637	-1.5	0.126	0.2	0.883
18–24 (ref)	229								
25–34	783	-0.1	0.913	−0.4	0.736	0.5	0.612	1.1	0.455
35–44	1329	0.0	0.991	−0.5	0.638	−0.2	0.838	−1.0	0.466
45–54	1469	0.1	0.929	0.0	0.965	0.0	0.974	0.6	0.684
55–64	1514	−0.1	0.857	−0.5	0.650	0.3	0.802	0.6	0.694
65–75	1429	−0.4	0.536	−1.0	0.358	−0.2	0.882	0.4	0.779
Sex									
Intercept		−0.1	0.736	−0.1	0.690	−1.1	<0.001	0.2	0.604
Male (ref)	3220								
Female	3533	0.1	0.825	0.2	0.585	−0.6	0.065	0.5	0.286
Education level									
Intercept		−0.1	0.552	−0.1	0.605	−1.5	<0.001	1.0	0.002
High (ref)	3724								
Middle	2329	0.1	0.611	0.3	0.495	0.4	0.254	−1.0	0.056
Low	700	0.2	0.643	0.3	0.609	−0.3	0.563	−2.0	0.014
Occupation status									
Intercept		0.1	0.591	0.2	0.475	−1.3	<0.001	0.6	0.057
Employed (ref)	3614								
Student	174	−0.3	0.679	−0.6	0.608	0.5	0.633	−0.8	0.618
Unemployed	1066	−0.2	0.490	−0.4	0.416	−0.3	0.489	−0.6	0.423
Retired	1547	−0.4	0.143	−0.7	0.120	−0.3	0.538	−0.3	0.599
Unable to work	352	0.5	0.345	1.2	0.132	−0.1	0.941	0.0	0.976
Income									
Intercept		0.4	0.091	0.5	0.180	−1.0	0.010	0.6	0.224
High (ref)	1436								
Middle	3275	−0.4	0.194	−0.4	0.380	−0.4	0.364	0.2	0.711
Low	1345	−1.3	<0.001	−1.6	0.004	−1.2	0.030	−1.1	0.153
Chronic disease status									
Intercept		−0.3	0.038	−0.5	0.034	−1.6	<0.001	0.2	0.560
No chronic disease (ref)	3881								
With chronic disease	2872	0.7	0.004	1.2	0.001	0.5	0.150	0.6	0.201
COVID-19 status									
Intercept		−0.1	0.427	−0.1	0.462	−1.4	<0.001	0.3	0.168
Not infected at T1 (ref)	6610								
Infected at T1	143	3.1	<0.001	6.2	<0.001	0.3	0.829	5.4	0.001
Living situation									
Intercept		−0.6	0.023	−1.0	0.016	−1.9	<0.001	−0.3	0.602
Not living alone (ref)	5469								
Living alone	1284	−0.7	0.016	−1.2	0.007	−0.6	0.197	−0.9	0.138
Country									
Intercept		0.8	0.078	1.0	0.114	−0.2	0.735	−5.3	<0.001
Greece(ref)	499								
Italy	1784	−0.3	0.532	−0.5	0.491	−0.7	0.371	5.7	<0.001
Netherlands	1143	−0.8	0.153	−0.8	0.307	−0.8	0.329	4.6	<0.001
UK	1448	−1.3	0.010	−1.7	0.027	−2.3	0.002	4.6	<0.001
US	1879	−1.1	0.022	−1.5	0.048	−1.4	0.052	9.1	<0.001

Note: In Table 2, univariate linear regression was used, and reference groups are marked with “(ref)”. We excluded data from 12 respondents residing in Greece from the analysis because of missing data on income.

**Table 3 jcm-11-06467-t003:** Multivariate analyses of respondent characteristics and the change in EQ-5D-5L transformed level sum scores, EQ-5D-5L transformed index, EQ VAS, and WHO-5 index.

		Change between T1 (April–May 2020) and T2 (May–June 2021)							
Characteristic	N	tEQ-5D Level Sum Score		tEQ-5D Index		EQ VAS		WHO-5 Index	
		Coeff.	*p*-Value	Coeff.	*p*-Value	Coeff.	*p*-Value	Coeff.	*p*-Value
Intercept		1.0	0.055	1.0	0.150	-0.2	0.735	-5.4	<0.001
Income									
High (ref)	1436								
Middle	3275	−0.5	0.154	−0.3	0.358				
Low	1345	−1.4	<0.001	−1.7	0.002				
Chronic disease status									
No chronic disease (ref)	3881								
With chronic disease	2872	0.8	0.001	1.2	<0.001				
COVID-19 status									
Not infected at T1 (ref)	6610								
Infected at T1	143	3.0	<0.001	6.1	<0.001			4.7	0.004
Country									
Greece (ref)	499								
Italy	1784	−0.2	0.623	−0.5	0.527	−0.7	0.371	5.6	<0.001
Netherlands	1143	−0.9	0.077	−1.1	0.149	−0.8	0.329	4.5	<0.001
UK	1448	−1.3	0.012	−1.7	0.026	−2.3	0.002	4.5	<0.001
US	1879	−1.3	0.011	−1.7	0.023	−1.4	0.052	8.9	<0.001
F-value		5.9	<0.001	5.9	<0.001	4.0	0.002	22.9	<0.001
R-square		0.008		0.009		0.002		0.017	

Note: In Table 3, multivariate linear regression was used, and reference groups are marked with “(ref)”. If a cell is empty, that variable is not in the final model. We excluded data from 12 respondents residing in Greece from the analysis because of missing data on income.

**Table 4 jcm-11-06467-t004:** Mean change in health-related quality of life and mental well-being by life event.

		Mean Change in Scores between T1 and T2			
Life Event	n	tEQ-5D-5L Level Sum Score	tEQ-5D-5L Index	EQ VAS	WHO-5
Number of chronic disease(s)					
Decreased	1202	**1.90**	**2.86**	**0.09**	**3.57**
Same	4672	**−0.003**	**0.02**	**−1.20**	**0.59**
Increased	891	**−2.81**	**−4.03**	**−4.53**	**−4.53**
COVID-19 status					
No (past) COVID-19 infection at T1 and T2	6442	**0.04**	0.07	−1.33	0.52
(past) COVID-19 infection at T1	27	**0.74**	1.26	−4.44	−4.74
(past) COVID-19 infection between T1 and T2	296	**−1.69**	−1.93	−2.86	−0.72
Vaccination status					
Received vaccine at T2	3945	−0.17	−0.22	−1.58	**1.36**
Not received vaccine at T2	2820	0.16	0.28	−1.17	**−0.83**
Change in work status *					
Gained job	218	1.35	1.91	−0.21	**2.44**
Kept job	3361	0.30	0.42	−1.08	**0.70**
Lost job	116	−1.77	−1.12	−4.59	**−3.10**
Remained unemployed	422	−0.37	−0.71	−0.98	**−0.59**
Change in income in past year (T1-T2)					
Improved	782	**−0.25**	−0.41	**−2.02**	**1.70**
Remained the same	4564	**0.16**	0.27	**−0.95**	**1.39**
Worsened	1322	**−0.60**	−0.77	**−2.61**	**−3.44**
Don’t know	97	**0.31**	0.39	**−1.55**	**−1.40**
Living situation					
Living with others at T1 and T2	5314	0.14	0.27	−1.30	0.68
Living alone at T1 and T2	1164	−0.52	−0.81	−1.69	−0.10
Living alone at T1 and with others at T2	124	−1.41	−2.28	−3.17	−1.81
Living with others at T1 and alone at T2	163	−1.17	−1.58	−1.53	−1.74

* Only those who were (self-)employed or unemployed at T1 and T2. Significant ddifferences in mean EQ-5D-5L transformed level sum scores, EQ-5D-5L transformed index, EQ VAS, and WHO-5 index are shown in bold.

## Data Availability

Data were collected and processed anonymously.

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
