# Peer review of "Health-Related Quality of Life and Mental Well-Being during the COVID-19 Pandemic in Five Countries: A One-Year Longitudinal Study"

_jcm, 2022, doi:10.3390/jcm11216467_

Round 1
Reviewer 1 Report
It is a very interesting study that has at least two great advantages: it is longitudinal and compares different countries. The methods and data analysis is rigorous and the conclusions are well supported by the results. But, I found it a bit difficult to read all the results and follow the discussions. In my opinion, the results are very complex and the discussions are more general. I would ask the authors to make more effort to discuss the various results in depth. For this reason, I think it is necessary a minor revision.
Please, check the text characters that are not uniform.
Furthermore, at line 80-81: “This study is part of the second wave of the POPulation health impact of the CORo- Navirus disease 2019 (COVID-19) pandemic (POPCORN) study.“, it should be specified which project it is, by whom it was promoted and financed and if there are data already published.
Author Response
Comment #1: It is a very interesting study that has at least two great advantages: it is longitudinal and compares different countries. The methods and data analysis is rigorous and the conclusions are well supported by the results.
Authors’ response: We thank the reviewer for this positive feedback.
Comment #2: But, I found it a bit difficult to read all the results and follow the discussions. In my opinion, the results are very complex and the discussions are more general. I would ask the authors to make more effort to discuss the various results in depth. For this reason, I think it is necessary a minor revision.
Authors' response: We have adjusted the discussion of the paper according to the reviewer's suggestion.
Comment #3: Please, check the text characters that are not uniform.
Authors’ response: Thank you for pointing this out. We have checked the text and changed the characters that were not uniform.
Comment #4: Furthermore, at line 80-81: “This study is part of the second wave of the POPulation health impact of the CORo- Navirus disease 2019 (COVID-19) pandemic (POPCORN) study.“, it should be specified which project it is, by whom it was promoted and financed and if there are data already published.
Authors’ response: We have included a description of the aims of the POPCORN study, funding of the study and a reference to the publication that detailed the results of the T1 data collection.
Reviewer 2 Report
Thank you for the opportunity to review the manuscript Health-related quality of life and mental well-being during the COVID-19 pandemic in five countries: a one-year longitudinal study. Generally speaking, the manuscript is well written and the methods are very good.
Strenghts: The paper has reviewed existing evidence and compares with the current ones. Also, the methods section clearly describes the objective of the study and its analysis to understand the results. The results are well described.
Still there are some statements which require more attention from the authors:
Lines 54-64 needs rephrasing.
Lines 320-330 needs rephrasing.
Author Response
Thank you for the opportunity to review the manuscript Health-related quality of life and mental well-being during the COVID-19 pandemic in five countries: a one-year longitudinal study. Generally speaking, the manuscript is well written and the methods are very good.
Strenghts: The paper has reviewed existing evidence and compares with the current ones. Also, the methods section clearly describes the objective of the study and its analysis to understand the results. The results are well described.
Authors’ response: We thank the reviewer for this positive feedback.
Still there are some statements which require more attention from the authors:
Comment #1: Lines 54-64 needs rephrasing.
Authors’ response: We have rephrased these lines.
Comment #2: Lines 320-330 needs rephrasing.
Authors’ response: We have rephrased these lines.
In addition, we have checked the text throughout the manuscript to improve the English language.
Reviewer 3 Report
This paper answers an interesting and relevant question and overall is an important contribution to the literature. The paper is relatively well executed, though I have several major concerns to highlight.
Major concerns:
· Overall the description of the data analysis should be improved.
· It is not clear how country effects were modelled/ taken into account. Were those modelled as fixed or random effects? Given the structure of your data - have you considered the use of a mixed-effects model?
· “We tested for differences in tEQ-5D-5L level sum score, tEQ-5D-5L index, EQ VAS and WHO-5 192 index by life event category with ANOVA and independent t-tests.” How did the structure of the data (i.e. data coming from different countries) was taken into account?
· This statement “Univariate linear regression analysis was then applied with as dependent the difference scores of the tEQ-5D-5L level sum score, tEQ-5D-5L index, EQ VAS,WHO-5 index, and as independents all determinants separately.” is confusing, suggest writing down the model. Particularly, it is difficult to comprehend “and as independents all determinants separately.”
· Kindly explain why you are using multivariate stepwise regression.
· Kindly consider using the non-parametric statistics: Wilcoxon signed-rank test and Kolmogorov–Smirnov tests in Table 4 as additional statistical tests.
Minor concerns:
· Statement “For analysis, the EQ-5D-5L level sum score and EQ-5D-5L index were transformed on a 168 0–100 scale (with 0 as worse anchor) in order to be comparable with the EQ VAS and 169 WHO-5 index.” requires a reference.
· In Table 2 include N. Details regarding the model should be included in Notes of the table.
· Kindly Include N in all the tables.
· The first statement of the abstract seems to be a little long.
· T1 and T2 are not defined in the abstract and might confused the reader.
· This statement is confusing “Once participating the data capture system did not allow for missing values.” Do you mean that there are no missing values in your sample?
Author Response
This paper answers an interesting and relevant question and overall is an important contribution to the literature. The paper is relatively well executed, though I have several major concerns to highlight.
Major concerns:
- Overall the description of the data analysis should be improved.
Authors’ response: We have changed the description of the data analysis. We think it is improved in clarity.
- It is not clear how country effects were modelled/ taken into account. Were those modelled as fixed or random effects? Given the structure of your data - have you considered the use of a mixed-effects model?
Authors’ response: We have now added country in the regression analyses. And results showed sometimes there were no significant between-country differences in terms of changes in HRQoL scores.
With regards to the choice of the statistical model: in general there could be several model choices, analysis of increment is one of the choices that compared evolutions between subjects correcting for difference at baseline. It is a very straightforward approach. We were primarily interested in the effects associated with observed determinants.
Our data are derived from a sample of the general population from only a few countries. We emphasised on the uniqueness of each country: country effects are treated as a characteristic. And we did not intend to generalize the estimates to other countries. In a mixed-effected model, we would’ve treated countries as the random effect. In addition, in our study, where we have a large sample and small number of countries (Nc>>C), the (marginal) estimates would be very similar, because all the variance used in mixed model estimation is from within, rather than between, countries[1].
[1] M. L. Bryan and S. P. Jenkins, "Multilevel modelling of country effects: A cautionary tale," European sociological review, vol. 32, no. 1, pp. 3-22, 2016.
- “We tested for differences in tEQ-5D-5L level sum score, tEQ-5D-5L index, EQ VAS and WHO-5 192 index by life event category with ANOVA and independent t-tests.” How did the structure of the data (i.e. data coming from different countries) was taken into account?
Authors’ response: Thank you for pointing this out. We have tested for differences of change in tEQ-5D-5L level sum score, tEQ-5D-5L index, EQ VAS, and WHO-5 index by life event category with the independent-samples Kruskal-Wallis test (changed to non-parametric rather than paramatric). Due to the small number of respondents for some of the categories if sub-divided by country (e.g. change in living situation), we decided to perform the analysis for the total sample rather than by country.
- This statement “Univariate linear regression analysis was then applied with as dependent the difference scores of the tEQ-5D-5L level sum score, tEQ-5D-5L index, EQ VAS,WHO-5 index, and as independents all determinants separately.” is confusing, suggest writing down the model. Particularly, it is difficult to comprehend “and as independents all determinants separately.”
Authors’ response: We have rephrased the sentence to “Univariate linear regression analysis was then applied for each outcome, namely change in tEQ-5D-5L level sum score, change in tEQ-5D-5L index, change in EQ VAS, and change in WHO-5 index. For each outcome, regression analysis was repeated for each baseline determinant as independent variable.”
- Kindly explain why you are using multivariate stepwise regression.
Authors’ response: Regression analysis was chosen because we were interested in the association between baseline characteristics and change in HRQoL and well-being scores. A stepwise method was used for consideration among as set of candidate variables. The end goal was to obtain the simplest best model.
- Kindly consider using the non-parametric statistics: Wilcoxon signed-rank test and Kolmogorov–Smirnovtests in Table 4 as additional statistical tests.
Authors’ response: Thank you for pointing this out. We have switched to non-parametric statistics, namely used the independent-samples Kruskal-Wallis test.
Minor concerns:
- Statement “For analysis, the EQ-5D-5L level sum score and EQ-5D-5L index were transformed on a 168 0–100 scale (with 0 as worse anchor) in order to be comparable with the EQ VAS and 169 WHO-5 index.” requires a reference.
Authors’ response: We have added a reference to the sentence, as the reviewer suggested.
- In Table 2 include N. Details regarding the model should be included in Notes of the table.
Authors’ response: We have included the N in table 2 and table 3. We have also added details of the model in the notes of the table.
- Kindly Include N in all the tables.
Authors’ response: THe N is now included in all tables.
- The first statement of the abstract seems to be a little long.
Authors’ response: We have removed the detailed description of recent life events from the first statement of the abstract.
- T1 and T2 are not defined in the abstract and might confused the reader.
Authors’ response: We have added the T1 and T2 period of data collection to the abstract.
- This statement is confusing “Once participating the data capture system did not allow for missing values.” Do you mean that there are no missing values in your sample?
Authors’ response: We have rephrased this sentence. It now reads: “Once participating, the data capture system did not allow for skipping questions or leaving questions unanswered. As a result, there were no missing values.”
Authors’ response: In addition, we have checked the text in the manuscript and made revisions in order to improve the English language and style.
Round 2
Reviewer 3 Report
I thank the authors for addressing the issues raised. The exposition of the paper has improved.
I have several comments which I hope will be useful:
I would be careful regarding “Many studies have suggested that HRQoL and mental well-being during the pandemic are persistently poorer compared to pre-pandemic [29, 38-40]. Our study shows that the effects of the pandemic have persisted, even among those that are not COVID-19 infected.”.
1. There are more recent studies which have found no change in relevant outcomes between pre and during the pandemic periods. For example a recent study by Bolbocean, C., Rhidenour,K. B., McCormack, M., Suter, B., & Holder, J. L.(2022). Resilience, and positive parenting in parents of children with syndromic autism and intellectual disability. Evidence from the impact of the COVID-19 pandemic on family’s quality of life and parent–child relationships. Autism Research, 1–18. https://onlinelibrary.wiley.com/doi/10.1002/aur.2825
2. The study did not collect data prior the pandemic. Thus, it is not possible to quantify HRQoL and other outcomes of respondents prior the pandemic and compare them to those collected during T1 and T2. Does your study really “shows that the effects of the pandemic have persisted”?
Please elaborate more on the strengths of the study. These are not well developed.
I do not think the last statement in Conclusion adds much “Collection of repeated HRQoL and mental well-being data among multi-country general population samples allows investigating HRQoL over time and cause-effect relationships between a wide range of possible determinants of health and HRQoL of the general population, both during and after the acute phase of COVID-19 pandemic.” I suggest removing it. Kindly develop further the Conclusion section.
Author Response
Authors' response: Thank you for reviewing our paper and for your valuable suggestions and comments. We believe that the manuscript has substantially gained in quality following these suggestions.
I have several comments which I hope will be useful:
Comment #1: I would be careful regarding “Many studies have suggested that HRQoL and mental well-being during the pandemic are persistently poorer compared to pre-pandemic [29, 38-40]. Our study shows that the effects of the pandemic have persisted, even among those that are not COVID-19 infected.”.
- There are more recent studies which have found no change in relevant outcomes between pre and during the pandemic periods. For example a recent study by Bolbocean, C., Rhidenour,K. B., McCormack, M., Suter, B., & Holder, J. L.(2022). Resilience, and positive parenting in parents of children with syndromic autism and intellectual disability. Evidence from the impact of the COVID-19 pandemic on family’s quality of life and parent–child relationships. Autism Research, 1–18. https://onlinelibrary.wiley.com/doi/10.1002/aur.2825
- The study did not collect data prior the pandemic. Thus, it is not possible to quantify HRQoL and other outcomes of respondents prior the pandemic and compare them to those collected during T1 and T2. Does your study really “shows that the effects of the pandemic have persisted”?
Authors’ response: We agree with the reviewer. Thank you for pointing this out. We have rephrased this and added the reference the reviewer used as an example.
The sentences now reads as follows: “Many studies have suggested that HRQoL and mental well-being during the pandemic in the general population are persistently poorer compared to pre-pandemic[29, 38-40], although in certain sub-populations no changes were observed [41]. Our study results may indicate that the effects of the pandemic have persisted, even among those that are not COVID-19 infected. However, it should be noted that in this study pre-COVID-19 HRQoL and mental well-being data are not available for the majority of respondents.”
Note: we wrote “for the majority of respondents”, since we have pre-COVID-19 HRQoL data for a small sub-sample (n~300) of the respondents.
Comment # 2: Please elaborate more on the strengths of the study. These are not well developed.
Authors’ response: We have added the following sentence on strengths of the study to the discussion of the paper: “The first wave of data were collected at the early phase of the COVID-19 pandemic, allowing for the comparison of HRQoL and mental well-being during times of acute disruption of daily life, social isolation, discontinuation of usual health care (first wave) and a later phase of the pandemic.”
Comment #2: I do not think the last statement in Conclusion adds much “Collection of repeated HRQoL and mental well-being data among multi-country general population samples allows investigating HRQoL over time and cause-effect relationships between a wide range of possible determinants of health and HRQoL of the general population, both during and after the acute phase of COVID-19 pandemic.” I suggest removing it. Kindly develop further the Conclusion section.
Authors’ response: We agree with the reviewer and have changed the conclusions of the paper. The adjusted conclusion is: “Our multi-country study on the course of HRQoL and mental well-being from the first wave of COVID-19 showed that slightly more respondents showed improved rather than deteriorated HRQoL and mental well-being, with some variation by outcome measure and country. The effect of negative life events on HRQoL and mental well-being was larger than the effect of positive life events.”